# An exploratory study of the impact of CT slice thickness and inter-rater variability on anatomical accuracy of malunited distal radius models and surgical guides for corrective osteotomy

Emilia Gryska[1,2]*, Per Fredrikson[1,2], Katleen Libberecht[1,2], Charlotte Stor Swinkels[1,2,3], Peter Axelsson[1,2], Anders Björkman[1,2]

**1** Department of Hand Surgery, Sahlgrenska University Hospital, Mölndal, Sweden, **2** Institute of Clinical Sciences, Sahlgrenska Academy, University of Gothenburg, Gothenburg, Sweden, **3** Department of Medical Physics and Biomedical Engineering, Sahlgrenska University Hospital, Gothenburg, Sweden

* emilia.gryska@gu.se

## Abstract

High-resolution CT images are essential in clinical practice to accurately replicate patient anatomy for 3D virtual surgical planning and designing patient-specific surgical guides. These technologies are commonly used in corrective osteotomy of the distal radius. This study evaluated how the virtual radius models and the surgical guides' surface that is in contact with the bone vary between experienced raters. Further, the discrepancies from the reference radius of surgical guides and radius models created from CT images with slice thicknesses larger than the reference standard of 0.625mm were assessed. Maximum overlap with radius model was measured for guides, and absolute average distance error was measured for radius models. The agreement between the lower-resolution guides surface and the raters' guide surface was evaluated. The average inter-rater guide surface overlap was -0.11mm [95% CI: -0.13–0.09]. The surface of surgical guides designed on CT images with a 1mm slice thickness deviated from the reference radius within the inter-rater range (0.03mm). For slice thicknesses of 1.25mm and 1.5mm, the average guide surface overlap was 0.12mm and 0.15mm, respectively. The average inter-rater radius surface variability was 0.03mm [95% CI: 0.025–0.035]. The discrepancy from the reference of all radius models created from CT images with a slice thickness larger than the reference slice thickness was notably larger than the inter-rater variability but, excluding one case, did not exceed 0.2mm. The results suggest that 1mm CT images are suitable for surgical guide design. While 1.25mm slices are commonly used for virtual planning in hand and forearm surgery, slices larger than 1mm may approach the limit of clinical acceptability. Discrepancies in radius models were below 1mm, likely below clinical relevance.

**Data Availability Statement:** All relevant data are within the paper and its Supporting Information files.

**Funding:** The study was financed by grants from the Swedish state under the agreement between the Swedish government and the county councils, the ALF-agreement (ALFGBG-966260). The funders had no role in study design, data collection and analysis, decision to publish, or preparation of the manuscript.

**Competing interests:** The authors declare that they have no competing interests.

# Introduction

3D virtual surgical planning (VSP) and patient-specific surgical guides (PSSGs) for the reconstruction of anatomy in complex skeletal deformities are becoming more common in hand and forearm surgery [1–3]. PSSGs are designed to fit a patient's bone perfectly in the guide's designed position. Therefore, PSSG design assumes that the virtual bone models used to design the guides exactly represent the patient's anatomy [4–6]. High-quality and high-resolution CT images are used in 3D VSP and PSSG design to fulfil this assumption. In hand and forearm surgery, companies offering 3D VSP and PSSG design request CT images with slice thicknesses between 0.5 and 1.25mm [7, 8]. The preference for high-resolution images often leads to additional dedicated CT exams, despite the availability of lower-resolution scans from routine clinical work-ups. This results in extra radiation exposure for patients and increased resource utilization.

Virtual bone models created from semiautomatic, expert segmentations are currently the gold standard in research and clinical practice. Although the patient-specific approach assumes that the virtual bone models exactly represent a patient's anatomy, discrepancies from the ground truth occur in practice. Even with high-resolution images, variations in image segmentation are expected between raters. Additionally, the accuracy of 3D-printed surgical guides is affected by printing errors. Despite these deviations, clinical outcomes of the patient-specific approach are still superior to the conventional method [1–3].

As 3D VSP and PSSG design hold the potential to become the gold standard for complex corrections in skeletal surgery, it is crucial to analyse the discrepancies in the virtual models created from the same images by various raters (inter-rater variability) and the CT image slice thickness.

Previous studies have assessed how increasing the reconstructed CT slice thickness influences the accuracy of virtual or printed anatomical models for mandible [9, 10], bovine vertebra [11], and skull models [12]. The results of these studies varied, suggesting the maximum slice thickness to be between 0.3mm [10] and 3mm [9]. No previous study, however, assessed the impact of increasing CT slice thickness on the accuracy of virtual radius models; whether these radius models can be used to design well-fitting PSSGs; and the inter-rater variability in models created from the standard CTs with low slice thickness.

The aim of this exploratory study was three-fold. First, we aimed to analyse the inter-rater surface variability of the typical contact area for surgical guides on the volar distal radius. Second, we aimed to analyse the surface discrepancy of surgical guides designed for radius models created from lower-resolution CT images (slice thickness 1.6–4 times larger than the reference 0.625mm) when fitted to the high-resolution reference radius model. Our final aim was to analyse the inter-rater surface variability of the entire radius and radius models created from the lower-resolution CT images. This exploration will lay the groundwork for ensuring that the chosen CT acquisition protocols guarantee adequate image quality while considering patients' safety and the efficient use of hospital resources.

# Materials and methods

CT images and VSP of twelve consecutive participants (11 women and one man, median age 61 years, range 21 to 73 years) scheduled for corrective osteotomy of the distal radius were available for this study. The data were collected with the approval of the Swedish Ethics Review Board (2021–01974). The patients' data were accessed between September 15th, 2023, and June 26th, 2024. The authors had access to information that could identify individual participants during and after data collection.

## Radiological images

All CT images used for the surgical planning and guide design were acquired according to the *Materialise* (NV Leuven, Belgium) protocol [7] with 0.39mm x 0.39mm pixel size and 0.625mm slice thickness on a *GE Discovery CT750 HD* scanner (GE Healthcare, Milwaukee, Wisconsin, USA). The original CT of all twelve participants were resliced to slice thickness equal to 1mm, 1.25mm, 1.5mm, 1.875mm, and 2.5mm: the images were converted from DICOM (Digital Imaging and Communications in Medicine, http://medical.nema.org/) to NIfTI (Neuroimaging Informatics Technology Initiative, https://nifti.nimh.nih.gov/) format using the *dcm2niix* tool [13] and resliced in Python (v3.10.13) using the *resample_img()* function with bicubic interpolation (*interpolation =*"*continuous*") from *nilearn* library (v0.10.2). The pixel size remained unchanged for all images. The resliced images were converted back to DICOM format in *3D Slicer* (v5.2.2, https://www.slicer.org/) so they could be opened by the *Mimics* (Materialise NV, Leuven, Belgium) software we use for segmentation and for generating virtual models.

## Virtual models

Virtual radius models were created in *Mimics* (Materialise NV, Leuven, Belgium) from CT images. First, each image was segmented: a radius mask was created using semiautomatic segmentation functions and manual editing. Then, the segmentation mask was converted to a 3D Part using the *Optimal quality* level option in *Mimics*. Further local manual smoothing of the 3D Part edges was done as necessary to ensure that the radius model outline followed the shape of the bone in the image. Each rendered bone model was saved as a Stereolithography (STL) file representing a virtual model's surface with a triangular mesh.

A surgical guide for a corrective osteotomy typically consists of a base that is shaped so that the guide is stable once positioned on the bone. The surface of the guide base that touches the bone perfectly matches the surface of the virtual radius model. The guide has openings for cutting the bone and pre-drilling the bone for screws to fix the plate. In this study, we were only interested in whether and how the guide surfaces designed based on lower-resolution radius models deviate from the reference radius model. Therefore, only a guide base for each radius model was created in 3-Matic and saved as an STL file. The workflow of creating the models is shown in Fig 1.

Reference models: three experienced raters (one surgeon and two clinical engineers) created reference radius models and surgical guide bases of the same shape, fitting these reference models for all 12 patients using the original CT images (slice thickness of 0.625mm).

"Lower-resolution" models: One rater created five radius- and five guide models—one pair for each resliced image—for all twelve participants. Fig 2 shows an example of a radius model at each resolution.

## Analysis

Our analysis was based on a part-to-part comparison of guide models to the reference radius models (guide-to-radius analysis). An additional analysis of the radius model (radius-to-radius) was also conducted for the whole radius and individually for the distal and proximal parts. Each radius model was split by a plane perpendicular to the long axis of the bone positioned directly below the proximal edge of the guide base. The Visualization Toolkit library (v9.2.6) [14] implemented in Python was used to import the STL models as *vtkPolyData*. The distances between the corresponding guide and reference radius mesh nodes were calculated using the *vtkDistancePolyDataFilter* function.

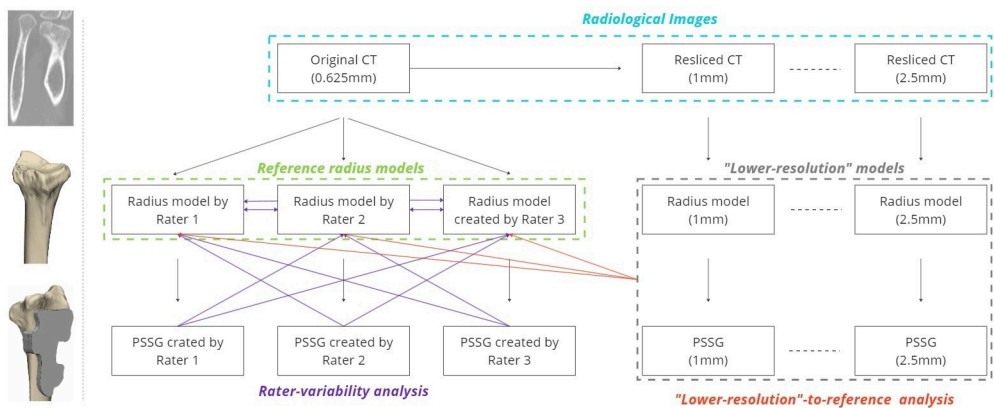

**Fig 1. The workflow shows how the models used in this study were created for one participant.** In short, from each CT image (top row of the diagram), a radius model (middle row) and a perfectly fitting surgical guide base model (bottom row of the diagram) were created. Three raters created radius and guide base models from the original CT images (left part of the diagram). One rater created radiuses and guide bases from resliced CT images (right part of the diagram). A sample of the CT, radius, and guide base models is shown on the left. This procedure was repeated for all twelve participants.

The guide-to-radius analysis measured the discrepancies between the guide surface and the radius surface in the guide's designed position. Specifically, we calculated the negative Hausdorff distance (HDF) for guide-to-radius comparison, which indicates the largest overlap between the reference radius model and the guide base surface. Such overlap could potentially lead to a worsened fit of the guide in real life. The absolute average distance error (ADE) was calculated for radius-to-radius comparisons. Both the negative HDF and absolute ADE were selected as the most relevant measures of surface discrepancy.

To examine the inter-rater guide (and radius) surface variability, we calculated the average negative HDF across the raters: each rater's guide models were compared to the other two rater's radius models (Fig 1., left part of the diagram). For the twelve participants and two comparisons for each of the three raters, we calculated the average and the 95% confidence interval (CI) of 72 (12x3x2) HDF values (and absolute ADE values for radius-to-radius comparison).

To analyse the impact of increasing the CT slice thickness on the surface discrepancy of guide models, each lower-resolution guide was compared to all three reference radius models and the average negative HDF of the three values was taken. (Fig 1, right part of the diagram).

Bland-Altman plots were constructed to assess the agreement in the average surface discrepancy from the reference (negative HDF) between the raters' guides and guides created for each lower-resolution radius model. The mean difference between the raters' and each lower-resolution model was compared to the average inter-rater variability. The normality of the differences was assessed with Q-Q plots (S1, S2 Tables in S1 File).

Finally, the models were visually analysed to determine whether any anatomical regions were particularly prone to deviations. The distance between models was visualised in FreeCAD (v0.21.1, https://www.freecad.org/).

## Results

The average negative HDF across all raters' guides was -0.11mm [95% CI: -0.13–0.09]. The Bland-Altman analysis (Fig 3) showed an increasing mean difference between the raters and each consecutive CT slice thickness model. Even though one outlier outside of the 95% CI was present for the differences between the raters and the models from CT with 1.875mm slice

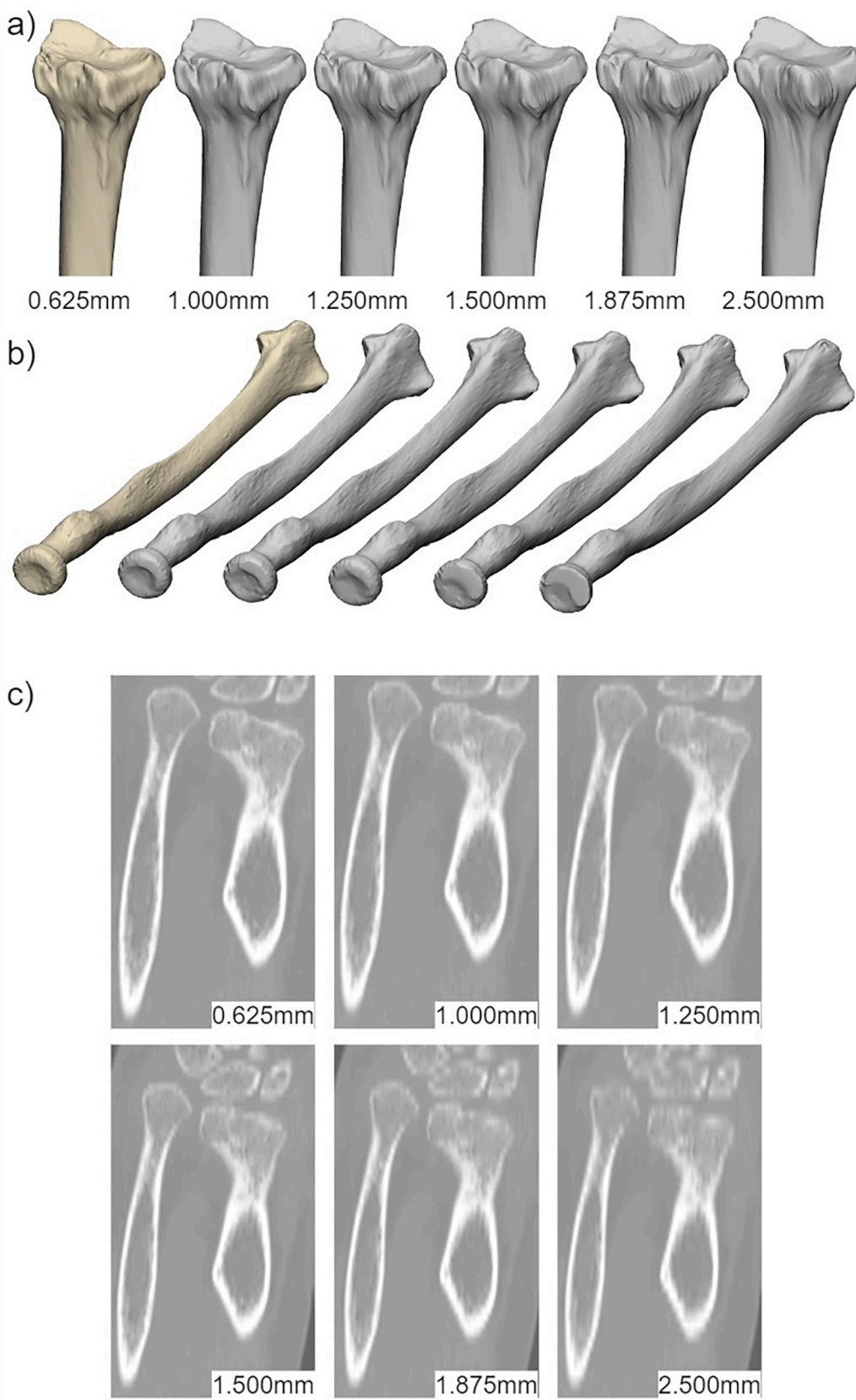

**Fig 2.** Two views (a and b) of one participant's radius models at each CT image slice thickness (c); the models are arranged left to right according to the slice thickness of the CT image used to generate the model. The leftmost model is the reference created on the original CT slice thickness of 0.625mm.

thickness, we assumed a normal distribution of the data based on the Q-Q plots (S1 Table in S1 File).

For CT slice thickness = 1mm, the mean difference from the raters was 0.03mm. Although the CI of the mean difference does not include the equality line (Fig 3A), the limits of agreement are within the average inter-rater variability of ±0.11mm. For CT slice thickness of 1.25mm and 1.5mm (Fig 3B and 3C), the mean difference was 0.12mm and 0.15, respectively. The lower CI bounds of both mean differences fell within the average inter-rater variability, indicating agreement on the clinically acceptable limit. For CT slice thickness of 1.785mm and 2.5mm, the mean difference and its CI interval exceeded the inter-rater discrepancy, indicating poor agreement (Fig 3D and 3E).

The average inter-rater absolute ADE for radius-to-radius comparison was 0.03mm [95% CI: 0.025–0.035]. For the distal radius, it was 0.025mm [95% CI: 0.023–0.027] and for the proximal part: 0.03mm [95% CI: 0.22–0.037]. Since we could not assume a normal distribution for the Bland-Altman analysis in two out of five groups (S2 Table in S1 File), we show the distribution of the absolute ADE for every pair of higher-resolution models compared to every rater's reference model, separately for the distal and proximal parts (Fig 4). The median absolute ADE is generally larger than the average inter-rater variability for all CT slice thicknesses and both the distal and proximal parts. Overall, we observed larger ADE values for the distal part than the proximal part. The median absolute ADE created from CT with a slice thickness of 1.25m and 1.5mm are comparable. Much larger absolute ADE and data spread are observed

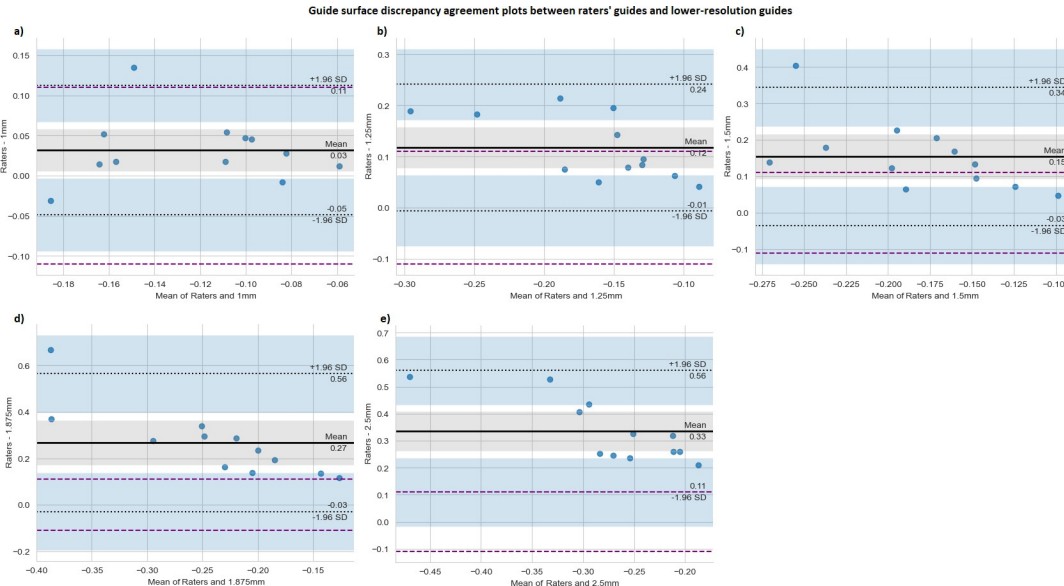

**Fig 3.** Bland-Altman plots assessing agreement of guide surface discrepancy from reference radius models between guides created by three raters on the original CT image with 0.625mm slice thickness and guides created from CT images with a) 1mm, b) 1.25mm, c) 1.5mm, d) 1.875mm, and e) 2.5mm slice thickness. The y-axis shows the difference between the average inter-rater surface discrepancy and the average discrepancy of a guide's surface created from lower-resolution CT from each reference model for each participant. The x-axis shows the mean of the average inter-rater discrepancy and the average discrepancy of the lower-resolution models. The grey area shows confidence intervals for the bias; the blue areas show the confidence intervals for the limits of agreement; the purple line indicates the ±average inter-rater discrepancy.

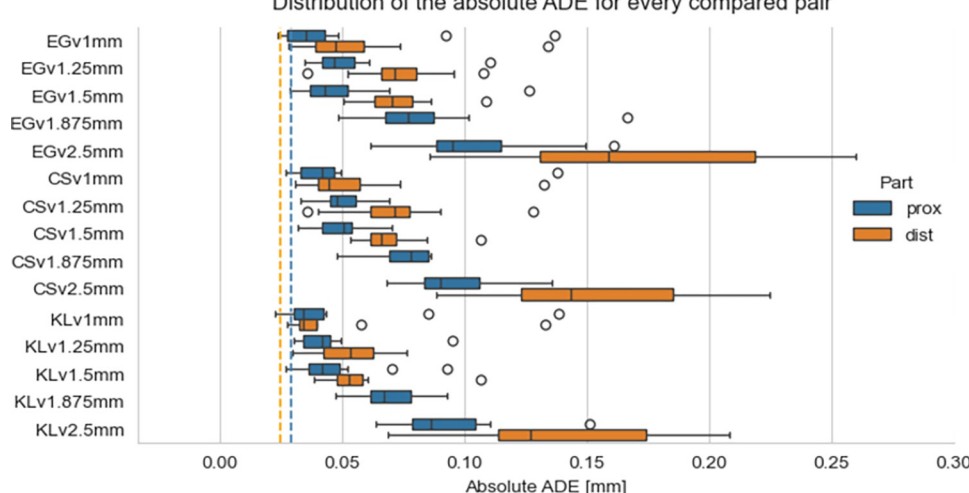

**Fig 4. Distribution of absolute ADE for radius models separated into the distal (dist) and proximal (prox) parts, created from CT images with a slice thickness of 1mm, 1.25mm, 1.5mm, 1.875mm, and 2.5mm compared to each rater's (EG, KL, CS) reference.** The dashed lines indicate the average inter-rater absolute ADE. One outlier of ADE > 0.65 for CSv2.5mm not shown.

for models created from CT images with a slice thickness of 1.875mm and 2.5mm, especially for the distal radius.

Visual analysis showed that the increment in slice thickness affected the accuracy of parts of the bone protruding along the long axis of the radius. This was also the direction along which the image resolution decreased (the CT images were also acquired axially). The deviation, however, did not seem to follow a consistent pattern. For example, Fig 5 shows how the radius models deviated from the reference.

The visual analysis of the guide-to-radius fit shows deviations in bone protrusions. Again, the pattern is not consistent as the slice thickness increases. In Fig 6, a deviation of -0.5mm for the guide model created on 1.825mm slice thickness can be observed around the volar ulnar corner of the radius. The negative distance indicates that the guide overlapped with the bone. A deviation in a similar spot is visible for a slice thickness of 2.5mm but in a positive direction. An inconsistent deviation was also observed on the wings of the guide that oscillates between positive and negative deviation for consecutive slice thickness within 0.1mm discrepancy (Fig 7).

## Discussion

In this exploratory study, we evaluated the inter-rater surface variability of distally malunited radii and PSSG designed for their corresponding radius models. Further, the impact of increasing the CT image slice thickness on the surface discrepancies of radius models and surgical guides designed for these models from gold-standard references was examined.

The surgical guides' average inter-rater surface variability (negative HDF) was -0.11mm [95% CI: -0.13–0.09]. We propose to use this average inter-rater variability as the clinically acceptable surface discrepancy. Based on this clinically acceptable limit, we can conclude that CT images with a 1mm slice thickness can be used for creating virtual radius models and designing surgical guides. For guides created from CT images with slice thicknesses of 1.25mm and 1.5mm, the surface discrepancies were found to be on the limit of being clinically acceptable. For radius models, the inter-rater absolute ADE was 0.03mm (0.03 for the proximal and 0.025 for the distal part).

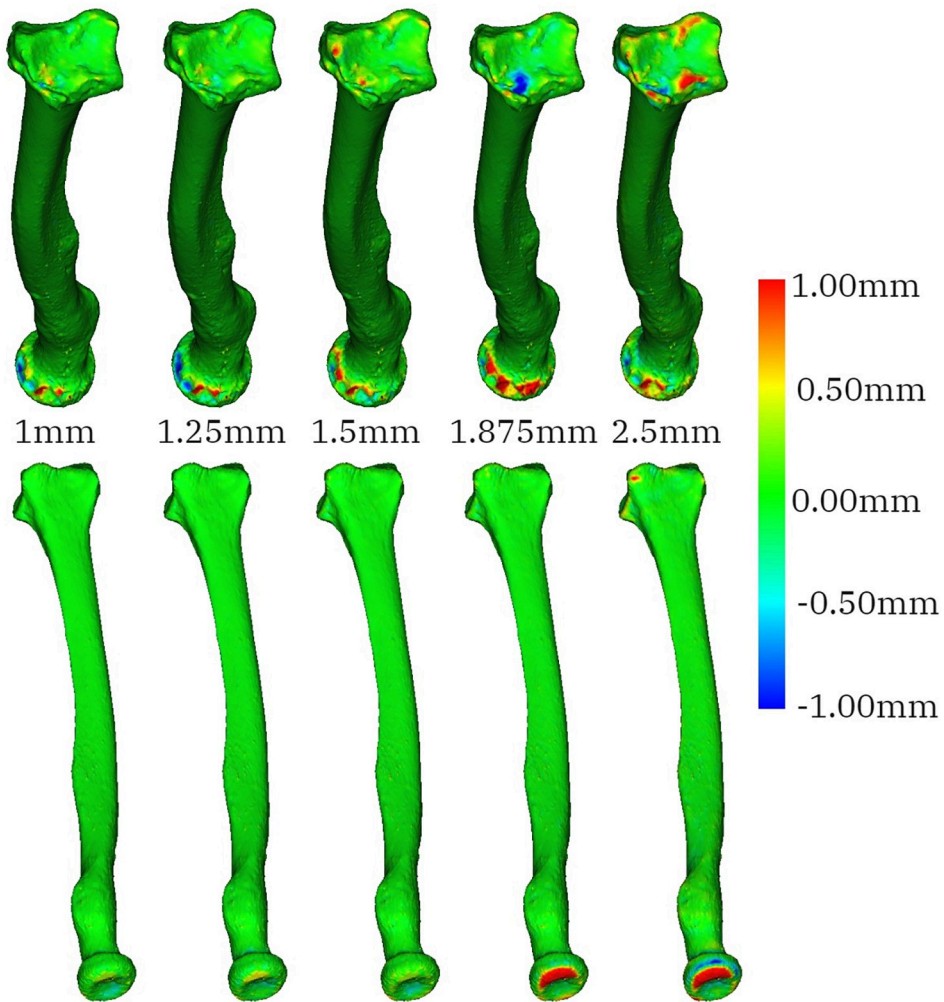

**Fig 5. Two views of the distance (shown in the colour bar) between the reference radius and lower-resolution radius models generated from CT images with slice thickness equal to 1mm, 1.25mm, 1.5mm, 1.875mm and 2.5mm for participant 1.** Discrepancies from the reference model can be noted in the head of the radius and in the most distal parts of the radius.

To our knowledge, this is the first study to assess the variability of gold-standard virtual radius models and PSSG surfaces across raters experienced in 3D VSP and PSSG design. The models created by any of the three raters can be considered the gold standard and have been used in clinical cases. Therefore, we decided to use all three as references. Since the gold standard is the best approximation of the ground truth given currently available imaging techniques, we assume that the variability in the gold standard is a clinically acceptable level of surface discrepancy. In this study, we used this variability as the limits of clinically acceptable surface discrepancy of PSSG designed for radius models created from CT images with varying slice thickness. Our results suggest that CT images with a slice thickness of 1mm can be used for PSSG design. Although CT images with a slice thickness of 1.25mm are used for 3D VSP and PSSG in hand and forearm surgery [8], our results suggest that the accuracy of PSSG created with these CTs may be at the limit of clinical acceptability.

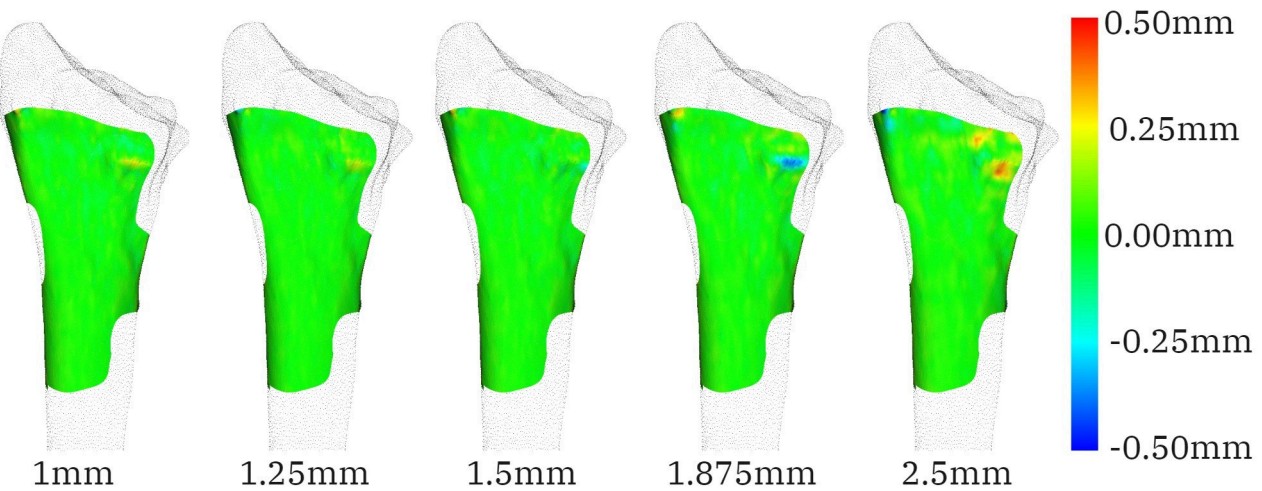

**Fig 6. The distance (indicated in the colour bar) between the reference radius and guide bases fitting radius models generated from CT images with slice thickness equal to 1mm, 1.25mm, 1.5mm, 1.875mm and 2.5mm for participant 3.** A deviation of 0.5mm is observed in opposing directions for the 1.875mm and 2.5mm models.

Corroborated by larger-scale studies, our results could be used to adjust CT acquisition and reconstruction parameters for 3D VSP and PSSG design. During acquisition, the parameter that can be adjusted to decrease the slice thickness is the helical pitch. Increasing the pitch will result in a lower resolution but also decreased radiation exposure and time of the examination. Once the data is acquired, the final image resolution can be adjusted during reconstruction. Decreasing image resolution during reconstruction will result in less noisy image. Lower-

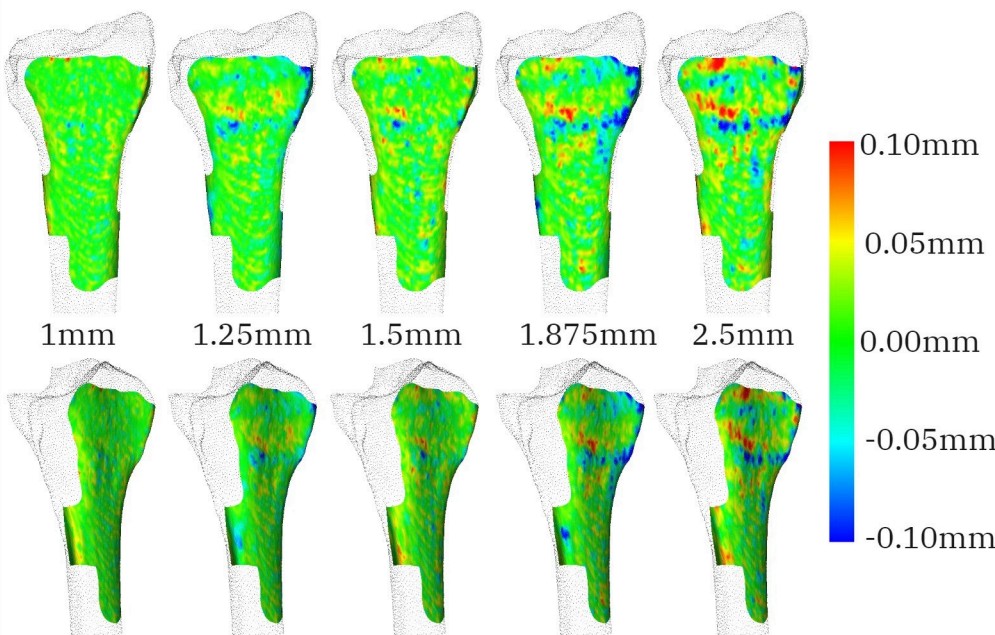

**Fig 7. Two views that show the distance within the 0.1mm range (indicated in the colour bar) between the reference radius model and guide bases fitting radius models generated from CT images with slice thickness equal to 1mm, 1.25mm, 1.5mm, 1.875mm and 2.5mm for participant 4.**

resolution images also take up less space. Even though this is rarely considered when optimising imaging protocols, virtual storage has a cost. Our and future results should guide medical physicists in designing new protocols to optimise resource use and patient safety. Previous studies that investigated the "docking robustness" [4, 5] of PSSG and developed a method to assess a guide's fit to the bone [6] assumed that the guide surface matches exactly that of the bone in its designed position. These studies also focused on modifying the shape of the guide base rather than assessing the accuracy of the models for which the guides are designed. Therefore, we could not use the previously proposed approach to assess how the guides designed for lower-resolution radius models fit the reference model, as the surface match assumption could not be met. While clinically acceptable discrepancy levels are known, we presume that measuring surface discrepancy is an alternative way to assess how well a guide will fit the bone. We used the negative HDF to measure guides' surface discrepancy as we needed a metric sensitive to localised overlaps with the bone. We suspect that the fit of the guide could worsen if such overlaps in virtual models are present.

Printing errors will also impact the fit of the actual guide to the bone. At our institute, the surgical guides are printed using selective laser sintering (SLS). The printing error reported in the literature for SLS varies significantly: Salmi et al. [15] reported an SLS print error of 0.79mm, while Msallem et al. [16]– 0.07mm. The impact of the printing errors on the accuracy of the guides designed with CT images with lower resolution must be tested on physical models or in a cadaver study. Such a study will also validate the surface discrepancy as a reliable measure of PSSG fit.

The secondary aim of our study was to compare radius models. The absolute ADE from the reference was discernibly larger than the inter-rater absolute ADE at every CT slice thickness level. Previous studies that assessed the accuracy of 3D (virtual and printed) anatomical models created at varying CT slice thicknesses focused on models of skulls [12] and mandibles [9, 10]. Ford and Decker [12] concluded that a CT slice thickness of 1.25mm should not be exceeded for accurate anatomical reconstruction of a skull. A similar threshold was suggested by Whyms et al. [10] in a study on mandible specimens. From our analysis, we can notice that the absolute ADE from the reference was relatively similar between models from CTs with 1.25mm and 1.5mm slice thickness. Visibly larger absolute ADE vas noted for models created from CTs with 1.875mm and 2.5mm slice thickness, especially for the distal radius. We chose the absolute ADE as the most relevant metric to assess radius model discrepancies, and we can see the expected influence of increasing the CT slice on the discrepancy; however, we suspect that the deviations in absolute ADE within 1mm are not clinically relevant [17, 18]. This indicates that absolute ADE and our study design may not be adequate to establish the largest CT slice thickness suitable for 3D VSP.

The study has several other limitations. First, we only had images, 3D VSPs, and PSSG designs available from twelve participants. Equivalence studies on a larger sample are necessary to conclude the largest CT slice thickness that can be safely used for 3D VSP and PSSG design. Furthermore, our cohort included only one man, and the median age of the participants was above 60 years old. The size variation of the assessed bones was also low in the cohort and therefore we could not assess whether the results depend on the bone size. Based on the limited, demographically homogeneous cohort, we also could not assess the impact of age or sex on the results. In our study, the lower-resolution images were created by reslicing existing images in only one direction. Therefore, our study could not account for the potential influence of noise on the images for varying slice thickness [19] nor the impact of changing the pixel size. The appearance of the images in our study could also be influenced by the interpolation method; however, a visual comparison between images interpolated with the bicubic and

linear functions (results not included) did not show any perceivable differences that could alter the results.

A larger study is necessary to perform equivalence tests of models for 3D VSP and PSSG design created from CT images with larger slice thickness to those currently used. Ideally, images and VSPs from multiple centres would be acquired to further study the impact of a CT machine on the images, and various raters on the segmentation. A more varied and larger cohort would further allow to investigate other factors that could influence the optimal imaging protocol, like age and age-related conditions, like arthritis, ethnicity, or sex, mentioned in the limitations section. We also suggest using a different metric than ADE to assess the suitability to the models for 3D VSP and normalise the results with respect to the bone volume, so the results are comparable across various cohorts. Intra-articular and diaphyseal malunions should also be considered. The results may also vary for other anatomies.

## Supporting information

**S1 File.**
(DOCX)

**S1 Raw data. Data used for this analysis are available as supporting information in data_raw.**
(ZIP)

## Author Contributions

**Conceptualization:** Emilia Gryska, Per Fredrikson.

**Data curation:** Emilia Gryska, Katleen Libberecht, Charlotte Stor Swinkels.

**Formal analysis:** Emilia Gryska.

**Funding acquisition:** Anders Björkman.

**Investigation:** Emilia Gryska.

**Methodology:** Emilia Gryska, Katleen Libberecht.

**Supervision:** Anders Björkman.

**Writing – original draft:** Emilia Gryska.

**Writing – review & editing:** Per Fredrikson, Katleen Libberecht, Charlotte Stor Swinkels, Peter Axelsson, Anders Björkman.

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
