## [Decision Letter · Decision Letter 0]

15 Sep 2024

PONE-D-24-32153An exploratory study of the impact of CT slice thickness and inter-rater variability on anatomical accuracy of malunited distal radius models and surgical guides for corrective osteotomy.PLOS ONE

Dear Dr. Gryska,

Thank you for submitting your manuscript to PLOS ONE. After careful consideration, we feel that it has merit but does not fully meet PLOS ONE’s publication criteria as it currently stands. Therefore, we invite you to submit a revised version of the manuscript that addresses the points raised during the review process.

We look forward to receiving your revised manuscript.

Kind regards,

Xiaohui Zhang

Academic Editor

PLOS ONE

“The study was financed by grants from the Swedish state under the agreement between the Swedish government and the county councils, the ALF-agreement (ALFGBG-966260).”

3. We are unable to open your Supporting Information file [data_raw.zip]. Please kindly revise as necessary and re-upload.

Reviewers' comments:

Reviewer's Responses to Questions

**Comments to the Author**

1. Is the manuscript technically sound, and do the data support the conclusions?

Reviewer #1: Yes

Reviewer #2: Yes

2. Has the statistical analysis been performed appropriately and rigorously? 

Reviewer #1: Yes

Reviewer #2: Yes

3. Have the authors made all data underlying the findings in their manuscript fully available?

Reviewer #1: Yes

Reviewer #2: Yes

4. Is the manuscript presented in an intelligible fashion and written in standard English?

Reviewer #1: Yes

Reviewer #2: Yes

5. Review Comments to the Author

Reviewer #1: The paper An exploratory study of the impact of CT slice thickness and inter-rater variability on

anatomical accuracy of malunited distal radius models and surgical guides for

corrective osteotomy is well written and documented. The aim was three-folded. Images are of high quality. Results and conclusions are clear. References are up to date.

Reviewer #2: The paper is very well-written, and the studies are well-designed. I have the following comments:

1. The Hausdorff distance and the ADE are absolute measures. It is true that absolute error is important in determining guide fit, however, it is possible that physically larger guides in larger patients will contribute more to the overall error in a study. That is, the composition of the patient cohort might affect the inference of the study. Hence, it might be useful to also report relative errors, i.e., absolute errors normalized by total volume, so that results are comparable across patient cohorts in future.

2. It would be useful to have a comment on the effects of image acquisition parameters that might affect the value of the optimal CT slice thickness discovered in this study.

3. Do the authors expect that the optimal CT slice thickness would be affected by demographic parameters, e.g., age, sex, given the anatomical differences associated with the same?

4. In the radius-radius comparison, it would be interesting to split the error into guide-contact region (distal radius and partial shaft) vs non-contact region (proximal radius and shaft). It is possible that given the high variations in small-scale structures in the radial head and neck, the clinically acceptable value discovered for the guide-contact region might be even lower than presented.

5. Please add a comment on factors that might contribute to different results in a large-scale study as compared to the exploratory study.

6. PLOS authors have the option to publish the peer review history of their article (what does this mean?). If published, this will include your full peer review and any attached files.

Reviewer #1: No

Reviewer #2: No

---

## [Author Response · Author response to Decision Letter 0]

17 Sep 2024

Dear Editor and Reviewers, 

We want to thank the reviewers for taking the time to review our manuscript; we appreciate the insightful and constructive comments. We believe that the manuscript was improved after we supplied the manuscript with the reviewer’s suggestions. In the following, we address all comments made by the editor and the reviewers. Changes to the manuscript are tracked, and comments are added in the manuscript to indicate which reviewer’s comment a given change refers to. 

Editor’s comments:

Authors’ response: We revised our submission and the manuscript to ensure that it meets PLOS ONE’s requirements. 

“The study was financed by grants from the Swedish state under the agreement between the Swedish government and the county councils, the ALF-agreement (ALFGBG-966260).”

Authors’ response: The funders had no role in the study, and the above statement should be added to the financial disclosure.

3. We are unable to open your Supporting Information file [data_raw.zip]. Please kindly revise as necessary and re-upload.

Authors’ response: We apologise for that. The original data are quite large and were provided in a .pkl format that is easy to open as a data frame using pandas library in Python. We now supplied the supporting information with the raw data available in text format so that it can be opened in other programs.

Authors’ response: We reviewed the reference list and formatted the citing style, so it meets PLOS ONE’s requirements

5. Review Comments to the Author

Reviewer #1: The paper An exploratory study of the impact of CT slice thickness and inter-rater variability on anatomical accuracy of malunited distal radius models and surgical guides for corrective osteotomy is well written and documented. The aim was three-folded. Images are of high quality. Results and conclusions are clear. References are up to date.

Authors’ response: We thank the reviewer for the positive feedback on our manuscript. 

Reviewer #2: The paper is very well-written, and the studies are well-designed. I have the following comments:

1. The Hausdorff distance and the ADE are absolute measures. It is true that absolute error is important in determining guide fit, however, it is possible that physically larger guides in larger patients will contribute more to the overall error in a study. That is, the composition of the patient cohort might affect the inference of the study. Hence, it might be useful to also report relative errors, i.e., absolute errors normalized by total volume, so that results are comparable across patient cohorts in future.

Authors’ response: Although this is an interesting thought, in case of the guide fit, we doubt that the size of the bone will influence the fit of the guide as measured by Hausdorff distance. We chose this metric as one localised overlap between the guide and the bone will have the same consequence for the guide fit, regardless of the size of the bone. Normalising the Hausdorff distance with bone volume could, however, incorrectly indicate a more serious negative effect on the guide fit in a smaller bone. 

Regarding the ADE, there could be a difference between larger and smaller bones, since relatively more detail is lost if the details are relatively smaller. However, as we stated in the discussion, we suspect that this metric may not be sensitive enough metric to establish the largest CT slice thickness suitable for 3D VSP. We agree, though, that the impact of bone size, e.g. in children or non-Caucasian populations should be accounted for in future studies. We mention the homogeneous bone size in our cohort as a limitation of this study:

“The size variation of the assessed bones was also low in the cohort and therefore we could not assess whether the results depend on the bone size.”

And in the last paragraph in response to Comment #5:

“We also suggest using a different metric than ADE to assess the suitability to the models for 3D VSP and normalise the results with respect to the bone volume, so the results are comparable across various cohorts.”

2. It would be useful to have a comment on the effects of image acquisition parameters that might affect the value of the optimal CT slice thickness discovered in this study.

Authors’ response: Thank you for this suggestion. We added a relevant paragraph in the Discussion:

“Corroborated by larger-scale studies, our results could be used to adjust CT acquisition and reconstruction parameters for 3D VSP and PSSG design. During acquisition, the parameter that can be adjusted to decrease the slice thickness is the helical pitch. Increasing the pitch will result in a lower resolution but also decreased radiation exposure and time of the examination. Once the data is acquired, the final image resolution can be adjusted during reconstruction. Decreasing image resolution during reconstruction will result in less noisy image. Lower-resolution images also take up less space. Even though this is rarely considered when optimising imaging protocols, virtual storage has a cost. Our and future results should guide medical physicists in designing new protocols to optimise resource use and patient safety.

3. Do the authors expect that the optimal CT slice thickness would be affected by demographic parameters, e.g., age, sex, given the anatomical differences associated with the same?

Authors’ response: We thank the reviewer for again bringing up a relevant point. We know from experience that image segmentation of arthritic and osteoporotic bone is difficult, and we suspect that the difficulty would increase with increased CT slice thickness. We know also that arthritis incidence increases with age and in older cohorts, women are more at risk. Given that our cohort mostly consisted of older women, we don’t have enough data to investigate the impact of the demographic parameters on the results. We added this to the limitations section in the manuscript:

“Furthermore, our cohort included only one man, and the median age of the participants was above 60 years old. The size variation of the assessed bones was also low in the cohort and therefore we could not assess whether the results depend on the bone size. Based on the limited, demographically homogeneous cohort, we also could not assess the impact of age or sex on the results. “

4. In the radius-radius comparison, it would be interesting to split the error into guide-contact region (distal radius and partial shaft) vs non-contact region (proximal radius and shaft). It is possible that given the high variations in small-scale structures in the radial head and neck, the clinically acceptable value discovered for the guide-contact region might be even lower than presented.

Authors’ response: We agree that this information would make our findings more interesting. We now added the results of radius models discrepancies split between the distal and the proximal parts in figure 4, as well as in the manuscript:

“An additional analysis of the radius model (radius-to-radius) was also conducted for the whole radius and individually for the distal and proximal parts. Each radius model was split by a plane perpendicular to the long axis of the bone positioned directly below the proximal edge of the guide base. “

“For the distal radius, it (ADE) was 0.025mm [95% CI: 0.023 -0.027] and for the proximal part: 0.03mm [95% CI: 0.22 -0.037].Since we could not assume a normal distribution for the Bland-Altman analysis in two out of five groups (S2 Table), we show the distribution of the absolute ADE for every pair of higher-resolution models compared to every rater’s reference model, separately for the distal and proximal parts (Fig 4). The median absolute ADE is generally larger than the average inter-rater variability for all CT slice thicknesses and both the distal and proximal parts. Overall, we observed larger ADE values for the distal part than the proximal part. The median absolute ADE created from CT with a slice thickness of 1.25m and 1.5mm are comparable. Much larger absolute ADE and data spread are observed for models created from CT images with a slice thickness of 1.875mm and 2.5mm, especially for the distal radius.” 

Fig 4. Distribution of absolute ADE for radius models separated into the distal (dist) and proximal (prox) parts, created from CT images with a slice thickness of 1mm, 1.25mm, 1.5mm, 1.875mm, and 2.5mm compared to each rater's (EG, KL, CS) reference. The dashed lines indicate the average inter-rater absolute ADE. One outlier of ADE > 0.65 for CSv2.5mm not shown.

5. Please add a comment on factors that might contribute to different results in a large-scale study as compared to the exploratory study.

Author’s response: We added the following paragraph in the Discussion:

“A larger study is necessary to perform equivalence tests of models for 3D VSP and PSSG design created from CT images with larger slice thickness to those currently used. Ideally, images and VSPs from multiple centres would be acquired to further study the impact of a CT machine on the images, and various raters on the segmentation. A more varied and larger cohort would further allow to investigate other factors that could influence the optimal imaging protocol, like age and age-related conditions, like arthritis, ethnicity, or sex, mentioned in the limitations section. We also suggest using a different metric than ADE to assess the suitability to the models for 3D VSP and normalise the results with respect to the bone volume, so the results are comparable across various cohorts. Intra-articular and diaphyseal malunions should also be considered. The results may also vary for other anatomies.”

---

## [Decision Letter · Decision Letter 1]

26 Sep 2024

An exploratory study of the impact of CT slice thickness and inter-rater variability on anatomical accuracy of malunited distal radius models and surgical guides for corrective osteotomy.

PONE-D-24-32153R1

Dear Dr. Gryska,

We’re pleased to inform you that your manuscript has been judged scientifically suitable for publication and will be formally accepted for publication once it meets all outstanding technical requirements.

Kind regards,

Xiaohui Zhang

Academic Editor

PLOS ONE

Additional Editor Comments (optional):

Reviewers' comments:

Reviewer's Responses to Questions

**Comments to the Author**

1. If the authors have adequately addressed your comments raised in a previous round of review and you feel that this manuscript is now acceptable for publication, you may indicate that here to bypass the “Comments to the Author” section, enter your conflict of interest statement in the “Confidential to Editor” section, and submit your "Accept" recommendation.

Reviewer #2: All comments have been addressed

2. Is the manuscript technically sound, and do the data support the conclusions?

Reviewer #2: Yes

3. Has the statistical analysis been performed appropriately and rigorously? 

Reviewer #2: Yes

4. Have the authors made all data underlying the findings in their manuscript fully available?

Reviewer #2: Yes

5. Is the manuscript presented in an intelligible fashion and written in standard English?

Reviewer #2: Yes

6. Review Comments to the Author

Reviewer #2: (No Response)

7. PLOS authors have the option to publish the peer review history of their article (what does this mean?). If published, this will include your full peer review and any attached files.

Reviewer #2: No

---

## [Editor Report · Acceptance letter]

1 Oct 2024

PONE-D-24-32153R1 

PLOS ONE

Dear Dr. Gryska, 

I'm pleased to inform you that your manuscript has been deemed suitable for publication in PLOS ONE. Congratulations! Your manuscript is now being handed over to our production team.

Kind regards, 

on behalf of

Dr. Xiaohui Zhang 

Academic Editor

PLOS ONE